# Paper Spray Mass Spectrometry on the Analysis of Phenolic Compounds in *Rhynchelytrum repens*: A Tropical Grass with Hypoglycemic Activity

**DOI:** 10.3390/plants10081617

**Published:** 2021-08-06

**Authors:** Cezar D. do Nascimento, Ana C. C. F. F. de Paula, Afonso H. de Oliveira Júnior, Henrique de O. P. Mendonça, Luisa del C. B. Reina, Rodinei Augusti, Rita de C. L. Figueiredo-Ribeiro, Júlio O. F. Melo

**Affiliations:** 1Department of Agrarian Sciences (DCA), Federal Institute of Education, Science and Technology of Minas Gerais (IFMG), Campus Bambuí, Rodovia Bambuí/Medeiros, km 05, Bambuí 38900-000, Brazil; cezar.diasn@gmail.com; 2Department of Exact and Biological Sciences (DECEB), Federal University of São João del-Rei (UFSJ), MG 424, km 47, Sete Lagoas 35701-970, Brazil; afonsohoj@gmail.com (A.H.d.O.J.); hp.quimico@hotmail.com (H.d.O.P.M.); 3Campus Sinop, Federal University of Mato Grosso, Av. Alexandre Ferronato, 1200—Res. Cidade Jardim, Sinop 78550-728, Brazil; luisabarrett@gmail.com; 4Department of Chemistry, Federal University of Minas Gerais (UFMG), Av. Pres. Antônio Carlos, 6627—Pampulha, Belo Horizonte 31270-901, Brazil; augusti.rodinei@gmail.com; 5Physiology and Biochemistry Section of Plants, Botanic Institute of São Paulo, Av. Miguel Stéfano, 3687—Agua Funda, São Paulo CEP 04301-902, Brazil; ritaclfribeiro@gmail.com

**Keywords:** PS-MS, bioactive compounds, Poaceae

## Abstract

The characterization of plant compounds with pharmacological activity is a field of great relevance in research and development. As such, identification techniques with the goal of developing new drugs or even validating the bioactive properties of extracts must be explored in order to further expand the knowledge of plant extract composition. Most works in this field employ HPLC, when exploring non-structural and cell wall carbohydrates from *Rhynchelytrum repens*. Phenolic compounds were studied by classical chromatography techniques and UV-vis spectrophotometry, with C-glycosylated flavonoids being detected but with no further details regarding the chemical structure of these compounds. In this work we employ paper spray ionization mass spectrometry (PS-MS) for the evaluation of the chemical profile of *R. repens* methanol extract. Positive ionization mode identified 15 compounds, belonging to flavonoids, fatty acids, and other classes of compounds; negative mode ionization was able to identify 20 compounds comprising the classes of quinic acids, stilbenes and flavonoids. PS-MS proved effective for the evaluation of *R. repens* extracts, making it possible to identify a total of thirty-five compounds. The bioactive properties attributed to *R. repens* were confirmed by the identification and characterization of compounds identified by PS-MS.

## 1. Introduction

A wide variety of plants have been traditionally and experimentally used in the treatment of diabetes mellitus, and the study of these species may result in potential new sources of antidiabetic drugs [1]. Hypoglycemic activity has been reported in several plants of the Poaceae family, such as *Rhynchelytrum repens*, a cosmopolitan grass found in tropical regions, and with a high level of adaptation to naturally low soil fertility and water availability. Popularly, the tea from this species, that is, its aqueous infusion, is used with the purpose of reducing symptoms in diabetes mellitus [2].

Studies carried out by our group with the aqueous extract of this plant have shown positive effects in decreasing glucose levels 1 h after administration. This effect was more pronounced after 3 and 4 h after administration when compared to control and other treatments such as the use of the supernatant residue, and in relation to the initial treatment time [3].

Structural carbohydrate such as β-glucans have been found in fractions of the extract that, when administered intraperitoneally, reduced the glucose levels in mice with streptozotocin-induced diabetes mellitus [4].

Oxidative stress, inflammation, and microbial infections are conditions that can be observed in the worsening of the general condition of diabetes mellitus. As such, another aspect of our research was to verify the presence of phenolic compounds, especially flavonoids and phenylpropanoids, since these compounds are known for their antioxidant, antimicrobial, anti-inflammatory, and antiviral activity [5,6,7]. Other bioactive properties include modulation of vascular homeostasis, as well as antidiabetic, anti-obesity, and anticancer effects [8,9].

Preliminary studies carried out with *Rhynchelytrum repens* extract using classical chromatographic techniques, showed the presence of phenolic compounds, especially flavonoids; however, the preliminary characterization was not sufficient for the identification of such compounds [3]. In this context, we propose the use of paper spray mass spectrometry (PS-MS), a methodology which has been increasingly used in exploring and fingerprinting chemical extracts from plant species [6]. PS-MS is an ambient ionization techniques, with high versatility, and presents a significant level of simplicity and low cost, standing out as a tool with high potential for advancing studies in this line of research [10,11].

Advantages of paper spray mass spectrometry range from shorter data acquisition time to stronger signal stability and higher replicability. The technique has shown to be able to compensate for limitations associated with other methodologies, overcoming disadvantages by enabling a quick acquisition of spectra in wide ranges with minimal need for sample preparation [12].

In light of the importance of phenolic compounds regarding bioactivity and the lack of data related to the identification of these compounds in *Rhynchelytrum repens*, this study employs PS-MS in the characterization of the methanolic extract of the species.

## 2. Results

### 2.1. Chemical Profile

Chemical profile spectra of the methanolic extract of *R. repens* in the positive and negative ionization modes identified 35 compounds. Positive ionization mode was able to identify 15 compounds, whereas negative ionization provided insight into 20 compounds in the phenolic extracts. The chemical profile is composed mostly of phenolic compounds (73%) comprising flavonoids (47%), other phenolic compounds (26%), and non-phenolic compounds, including sugars and carotenoids (27%).

### 2.2. Positive Ionization Mode

Figure 1 presents the spectrum generated from the methanolic extract of an *R. repens* sample in positive ionization mode.

Tentative identification of compounds using PS-MS in the positive ionization mode distinguished four chemical classes. Flavonoids represented most of the identified compounds, followed by phenylpropanoids as shown in Table 1.

The positive ionization method was applied in order to classify phenolic compounds and identify peaks based on their mass-to-charge ratio. Fragmentation of precursor ions enabled their identification based on their unique fragmentation patterns [13].

Phenolic compounds identified in the positive ionization mode were calycosin, identified as [M + H]^+^ ion at *m*/*z* 285; luteolin-8-C-(rhamno-sylglucoside) was identified as the [M + H]^+^ ion detected at *m*/*z* 595; chrysoeriol 7-rutinoside, diosmin, and rutin were detected as the [M + H]^+^ ions at 608, 609, and 611 respectively. Other flavonoids also detected were tricin-7-O-neohesperidoside and tricin7-O-deoxyhexosyl-glucuronide as the [M + H]^+^ ions at 639 and 653 respectively.

### 2.3. Negative Ionization Mode

A typical mass spectrum generated from analyzing the methanolic extract of *R. repens* in the negative ionization mode is shown in Figure 2. PS (-)-MS enabled the identification of a total of twenty compounds employing the tentative identification method with expected mass-to-charge ratios compared to data available in the literature.

Flavonoids represent a group of polyphenolic compounds with diverse chemical structure and properties found in plant species, with more than nine thousand different compounds described for this class [16]. Out of all identified compounds in this study, eighteen were classified as flavonoids, comprising a total of 85.71% from identified compounds detected by negative ionization.

In *R. repens*, flavonoids were represented by tricetin trimethyl ether as the [M − H]^−^ ion at *m*/*z* 343, as well as the detected [M − H]^−^ ions at *m*/*z* 417, 431, 445, and 461 identified as 6-C-pentosyl luteolin, apigenin-6-C-hexoside, swertisin, isoorientin, and isoscoparin, respectively. These compounds represent, however, only a fraction of noteworthy phenols in the species extracts. Table 2 presents the full list of tentatively identified compounds using PS-MS on negative ionization mode.

## 3. Discussion

Polyphenolic compounds are widely distributed in nature, being found in vegetables, fruits, and medicinal plants with remarkable biological effects and medicinal properties. Plants of the Poaceae family are economically very important representing some of the largest crops for human and animal consumption. Studies of phenolic compounds in Poaceae have been extensively carried out on sugar cane (*Saccharum officinarum* L.), an important species of this family, cultivated for the purpose of obtaining sugar and ethanol, with the latter widely used as biofuel in Brazil.

Works with sugar cane show high phenolic content and high antioxidant activity in the extracts [21]. Among sugarcane phenolics, flavonoids were specially detected in the form of methoxylated, C- and O- glycosylated flavones [14].

In contrast to sugarcane, with a production cost in the 2019/2020 crop of around 1600 USD per hectare, *Rhynchelytrum repens*, a spontaneous species commonly found associated with other crops, has few published studies. Resources are scarce or nonexistent in the scientific literature for this species. Work characterizing *R. repens* compounds employing HPLC techniques is limited to works carried out by our group on non-structural and cell wall carbohydrates [2,3]. This is no different when it comes to the characterization of its phenolic compounds.

Note that identifying the ions detected in the mass spectra is not an essential condition in differentiating the grass species evaluated herein as phylogeny presents the fact that *R. repens* belongs to the same subfamily as *Saccharum* spp. (sugarcane) within the PACMAD clade, in the subfamily Panicoideae, and that both share the C4 metabolism pathways [22]. C4 metabolism is typical of tropical grass species within this subfamily, as well as at the tribe and subtribe level for both Andropogoneae and Paniceae, which makes it possible to infer that the compounds earlier identified in sugarcane are the same as those found in *R. repens* due to their shared photosynthetic processes and by-products [23,24,25].

Furthermore, we also observe that the metabolic pathways for both species result in the same compounds which later take part in the synthesis of phenolic compounds as secondary metabolites [26]. By using this hypothesis, we have tentatively identified the compounds found in *R. repens* as listed in Table 1 and Table 2 which also present the sources from which we gathered the necessary information to ascertain their identity.

This grass species stands out as an invasive species to several economically important crops such as soy, beans, pineapple, rice, and sugarcane in different stages of the crop cycle. In South Florida, studies were carried out to test the invasive potential of *R. repens* in natural areas over native species and found that high densities of this species were associated with a significant reduction in the diversity of native species, showing a strong negative correlation with the diversity of native species [27]. However, here we highlight its quality as a plant rich in beta glucans, with hypoglycemic action and also rich in phenolics as detected by our findings through analyses employing PS-MS.

Among other phenolic compounds, 18 compounds were identified in the methanolic extract classified as flavonoids, revealing the range of compounds with antioxidant potential present in *R. repens*. In addition to that, it confirms preliminary results obtained from classical techniques such as TLC and hydroalcoholic extractions and resistance to hydrolysis with HCL 1 N [28]. As a matter of fact, it was possible to identify, with paper spray mass spectrometry, C-glycosylated flavonoids such as 6-C-pentosyl luteolin, apigenin-6-C-hexoside, and luteolin-8-C-(rhamnosyl-glucoside).

Tricin and luteolin 6,8-di-C-glucoside have been shown to possess a potent free- radical trapping activity, being considered a natural trapping agent for these radicals. Tricin-7-O-(200-a-L-rhamnopyranosyl)-a-D-galacturonide was also detected in *R. repens* with a reported antinociceptive activity [29,30,31]

Research of the main components present in the aqueous extract of this plant is currently being carried out by our group to further expand the knowledge of this species’ potential. Identification of such a wide range of *Rhynchelytrum repens* phenolic compounds reinforces the importance of this species, which should be the subject of further investigation.

## 4. Materials and Methods

### 4.1. Plant Samples

Samples of *Rhynchelytrum repens* were harvested manually with the inflorescence in order to facilitate identification, in the municipality of Bambuí, State of Minas Gerais, Brazil, at the geographic coordinates 20°1′17″ S 45°57′39″ W. The plant material was cleaned by removal of the root and inflorescence. For the experiment, only the aerial part comprising the leaves and stems were used. Specimens were collected during the flowering stage in the month of March (Southern Hemisphere summer), to properly identify the species and avoid sample contamination with other Gramineae species which could otherwise be mistaken for *R. repens* due to morphological similarities.

The samples were then kept at 60 °C for 72 h and then ground with mortar and pestle. The powder obtained from grinding was placed in glass containers and wrapped in aluminum foil for protection from light and kept at room temperature in the Laboratory of Plant Physiology, Department of Agrarian Sciences of the Federal Institute of Minas Gerais—IFMG—Campus Bambuí.

### 4.2. Methanolic Extract

The powder obtained from *R. repens* was used to prepare samples of 0.50 g to which 10.0 mL of methanol/water solution (50:50, *v*/*v*) was added. The samples were centrifuged for 20 s in a vortex mixer, and then kept in incubation for 24 h in the dark and at room temperature. After the incubation process, the samples were centrifuged for 15 min at 4 °C and 15,000× *g* rotation. The supernatant was transferred to a 2.5 mL Eppendorf microtube and when not being analyzed, kept at −18 °C in cold storage [26].

### 4.3. Paper Spray Mass Spectrometry

The analysis of the chemical profile of the samples was carried out using an LCQ fleet mass spectrometer (Thermo Scientific, San Jose, CA, USA) equipped with an ambient ionization source by paper spray. Spectra were acquired in triplicates for both positive and negative ionization modes [31]. For the analysis, 2 µL of the samples and 40 µL of methanol were applied to chromatographic paper attached to the equipment, after which the voltage source was connected for data acquisition.

Instrumental conditions of the analyses were: voltage of the PS-MS source equal to +4 kV (positive ionization mode) and −3 kV (negative ionization mode); capillary voltage of 40 V; transfer tube temperature at 275 °C; tube lens voltage of 120 V; mass-to-charge range from 100 to 1000 *m*/*z* in both positive and negative modes.

Ions were fragmented using collision energies of 15 to 45 eV. For the tentative identification of compounds, the instrumental signals obtained were compared to the mass-to-charge ratios found in the literature, followed by the fragmentation by sequential mass spectrometry in order to confirm the identification of each compound.

## 5. Conclusions

Paper spray mass spectrometry proved to be a potent tool for the characterization and determination of the quality of *R. repens* extracts because it is a simple, relatively fast and extremely efficient technique for determining the chemical profile. It enables the identification of a range of molecules distributed among different classes with distinct and synergistic properties. This can contribute to the better formulation of products derived from *R. repens*, providing antioxidant, anti-inflammatory, and stimulating effects, which can be explored pharmaceutically for the development of antidiabetic drugs.

## Figures and Tables

**Figure 1 plants-10-01617-f001:**
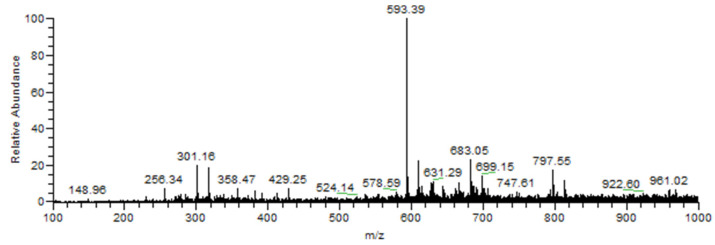
PS-MS spectrogram of *R. repens* extract in positive ionization mode.

**Figure 2 plants-10-01617-f002:**
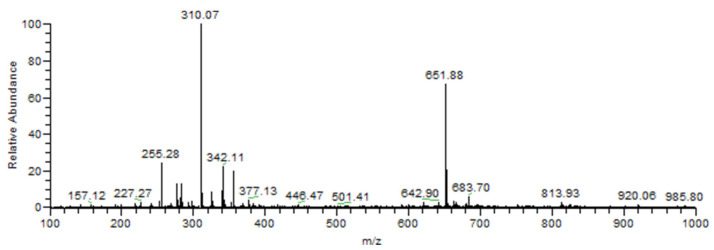
PS-MS spectrogram of *R. repens* extract on negative ionization mode.

**Table 1 plants-10-01617-t001:** Ion identification using PS(+)-MS.

Peak N.	Tentative Identification ^1^	*m*/*z*	MS/MS	Reference
1	hydroxy palmitic acyl amide	272	-	Gomes et al. (2020)
2	octadecatetraenoic acid	277	-	Alves et al. (2020
3	octadecatrienoic acid	279	-	Gomes et al. (2020)
4	calycosin	285	276, 267, 240	Silva et al. (2019)
5	hydroxy-octadecadienoic acid	297	-	Gomes et al. (2020)
6	epoxy-hydroxy-octadecadienoic acid	311	-	Alves et al. (2020)
7	docosenoic acyl amide	338	-	Figueirinha et al. (2008)
8	tetrahydroxycholanoic acid	425	-	de Oliveira Junior et al. (2020)
9	primulaverin	475	457, 433, 401	de Oliveira Junior et al. (2020))
10	luteolin-8-C-(rhamno-sylglucoside)	595	-	Figueirinha et al. (2008)
11	chrysoeriol 7-rutinoside	608	599, 594, 468	Campelo et al. (2020)
12	diosmin	609	-	Figueirinha et al. (2008)
13	rutin	611	618, 606, 602	Zhao (2018); Gomes et al. (2020)
14	tricin-7-O-neohesperidoside	639	-	Alves et al. (2020)
15	tricin7-O-deoxyhexosyl-glucuronide	653	-	Alves et al. (2020)

^1^ Identification based on expected mass signals [13,14,15,16,17,18,19,20,21,22,23,24,25,26,27,28,29,30,31,32,33].

**Table 2 plants-10-01617-t002:** Ion identification using PS (-)-MS.

Peak N.	Tentative identification ^1^	*m*/*z*	MS/MS	Reference
1	dicaffeoylquinic acid	515	497, 441, 383	Gomes et al. (2020)
2	(e)-resveratrol-3-O-glucopyranosyl-(1→6)-glucopyranoside	551	533, 523, 508, 389	Figueirinha et al. (2008)
3	tricetin trimethyl ether	343	325, 311, 299	Alves et al. (2020)
4	6-C-pentosyl luteolin	417	399, 290, 269, 209	Figueirinha et al. (2008)
5	apigenin-6-C-hexoside	431	370, 361, 318	Alves et al. (2020)
6	swertisin	445	207	Alves et al. (2020)
7	isoorientin	447	429, 419, 330, 163	Figueirinha et al. (2008)
8	isoscoparin	461	425, 407, 363	Figueirinha et al. (2008)
9	4′,5′-Dimethyl-luteolin-8-C-glucoside	475	453, 376, 283	de Oliveira Junior et al. (2020)
10	isorhamnetin-3-O-glucoside	477	449, 331, 263	de Oliveira Junior et al. (2020)
11	crysoeriol-7-O-malonylhexoside	547	529, 503, 277	Han et al. (2020)
12	6-C-pentosyl-8-C-pentosyl luteolin	549	549, 531, 505, 489	de Oliveira Junior et al. (2020)
13	schaftoside	563	519, 503, 490, 443, 413, 383	Figueirinha et al. (2008)
14	apigenin-6-C-hexoside-8-O-deoxyhexoside	577	504, 299, 277	Han et al. (2020)
15	6-C-hexosyl-8-C-hexosyl apigenin	579	561, 551, 535, 519, 489, 459, 399, 279	Figueirinha et al. (2008)
16	apigenin-di-C-hexoside	593	575, 456, 285	Han et al. (2020)
17	crysoeriol-6-C-hexoside-8-C-deoxyhexoside	607	563, 519, 487	de Oliveira Junior et al. (2020)
18	tricin-7-O-neohesperidoside	637	605, 561, 456	Alves et al. (2020)
19	tricin-7-O-(2″-α-L-rhamnopyranosyl)-α-D-galacturonide	651	591, 572, 565	Alves et al. (2020)
20	dicaffeoylquinic acid glycoside	677	659, 561, 550, 515	Gomes et al. (2020)

^1^ Identification based on expected mass signals [13,14,15,16,17,18,19,20,21,22,23,24,25,26,27,28,29,30,31,32,33].

## Data Availability

All data reported in this study are available on open repositories cited in sequence of presentation in the paper.

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
