# Peer review of "Paper Spray Mass Spectrometry on the Analysis of Phenolic Compounds in Rhynchelytrum repens: A Tropical Grass with Hypoglycemic Activity"

_plants, 2021, doi:10.3390/plants10081617_

Round 1
Reviewer 1 Report
The manuscript, "Paper spray mass spectrometry on the analysis of phenolic compounds in Rhynelytrum repens: A tropical grass with hypoglycemic activity", provides a communication of the implementation of the technique to the grass for phenolic determination. While the article is interesting it will be most focused to researchers using paper spray MS or studying Rhynchelytrum repens. Apart from this the manuscript is well written and provides sufficient detail to replicate the study or convert the study to other species. I therefore recommend the paper for publication.
Author Response
We revised the English language and style such as indicated below.
Reviewer 2 Report
- What month in which phase of growth were the plant harvested.
- Why inflorescence were not used for studied but only leaves and stems?
- phenolic compounds are better extracted at higher methanol concentrations, e.g. 70% or 80%
Author Response
Response to Reviewer 2 Comments
Point 1: What month in which phase of growth were the plant harvested
Response 1 - Specimens were collected during the flowering stage in the month of March, (Southern Hemisphere summer), so as to properly identify the species and avoid sample contamination with other Gramineae species which could otherwise be mistaken for R. repens due to morphological similarities.
Point 2 - Why inflorescence were not used for studied but only leaves and stems?
Response 2 - It is more common to prepare the tea of Rhynchelytrum repens without the inflorescence, and the extracts our group worked on considering the hypoglycemic activity also took in consideration previous knowledge and information provided by Dr Tatiana Sendulski who suggested this specific use of aerial parts.
Point 3 - Phenolic compounds are better extracted at higher methanol concentrations, e.g. 70% or 80%
Response 3 - We agree with the arguments brought forth by the reviewer when it comes to the extraction of phenolic compounds, as polar the use solvents such as methanol at concentrations of 70% and 80% are common, when considering only the extraction of phenolic compounds. However, our initial work considered the identification of compounds found in other chemical classes as well, not only restricting the protocol to the extraction of phenolic compounds, but to a broader range of compounds and classes. As such, we used the methodology as described in the manuscript, which provided satisfactory results.